# Knowledge of Breastfeeding Mothers Regarding Caries Prevention in Toddlers

**DOI:** 10.3390/children10010136

**Published:** 2023-01-10

**Authors:** Johnny Kharouba, Shaden Mansour, Tal Ratson, Sarit Naishlos, Gina Weissman, Sigalit Blumer

**Affiliations:** Department of Pediatric Dentistry, The Maurice and Gabriela Goldschleger School of Dental Medicine, The Faculty of Medicine, Tel Aviv University, Tel Aviv 6997801, Israel

**Keywords:** breastfeeding, knowledge, dental caries, infants, toddlers

## Abstract

Mothers’ awareness regarding the risk factors for the development of early childhood caries is crucial. The current study aims to examine the knowledge of breastfeeding mothers about their baby’s dental health and prevention of ECC while comparing primiparous mothers to multiparous mothers. A total of 165 mothers aged 20–49 y/o participated in the study by completing questionnaires that assessed the knowledge and attitudes of mothers toward their infants’ oral health. Results showed that (1) mothers were found to be highly knowledgeable regarding the oral hygiene of their infants and the recommended breastfeeding best practices (71%); (2) mothers with lower education showed poor knowledge regarding the recommended practices of infant oral health; (3) a large proportion of the mothers in the sample (62%) reported that they usually tasted the food before giving it to their baby, in a way they could transmit bacteria to infants; (4) most of the mothers (68%) were not aware that their dental health during pregnancy affects the infants’ dental health; and (5) multiparous mothers were more knowledgeable regarding artificial baby milk composition (96%) in comparison with mothers with only a single child (60%). According to the results, there is a need to improve the knowledge of breastfeeding mothers, especially mothers who have one child and mothers with a lower education about their baby’s dental health. The results of this study shed light on the knowledge of breastfeeding mothers on this important topic and could serve policymakers to improve practices toward advancing better oral health for infants, without sacrificing the benefits of breastfeeding, which are so crucial for infant health and development.

## 1. Introduction

### 1.1. The Benefits of Breastfeeding on Baby’s Health

Breastfeeding is widely acknowledged as the normal and superior infant feeding method due to its associated health benefits, both for the infant and the mother. Extensive evidence indicates that breastfeeding is vital to ensure the healthy growth and development of infants and children [1,2,3]. Epidemiological studies show that breast milk and breastfeeding improve the overall health, provide nutrition, psychological, social, and economic development, and provide environmental benefits. Breast milk also significantly reduces the risk of developing acute and chronic diseases [4]. Based on 113 studies, long-term breastfeeding is associated with a 26% reduction in obesity, and a decrease of 35% in type 2 diabetes. Moreover, breastfeeding is consistently associated with higher performance on intelligence tests in both children and adolescents [5].

Beyond the short-term benefits of breastfeeding, there are other long-term benefits. Breastfeeding reduces the risk of developing high cholesterol and high blood pressure later in life and is also associated with lower mortality and morbidity rates than infections in the infant’s gastrointestinal tract and reduces the risk of celiac disease and asthma [6]. Studies have also found that breastfeeding reduces infections in children (such as ear infections and pneumonia), reduces immune diseases (such as leukemia and inflammatory bowel disease), and reduces the risk of death in the first year of a baby’s life [7].

Leading health organizations such as the American Academy of Family Physicians, and the World Health Organization (WHO) recommend that infants be exclusively breastfed for about the first 6 months, and then combine continued breastfeeding with solid foods. Artificial baby milk (infant formula) should only be given to the baby if they have not been breastfed before one year of age [8,9,10,11,12].

### 1.2. Breastfeeding Milk and Oral Hygiene

Studies show a relationship between poor maternal oral hygiene and the health of the fetus [13,14]. Moreover, studies demonstrate a link between periodontal disease and adverse consequences in pregnancy such as preterm birth, low birth weight babies, and preeclampsia [15,16]. Mothers with poor oral hygiene and high levels of cariogenic oral bacteria are at the highest risk of infecting their children and increasing the risk for tooth decay in their children at an early age [15].

Dental caries is the most common chronic infectious disease in childhood. The etiology of dental caries is assumed to be an interaction of bacteria, mainly Streptococcus mutans, and sweet foods that are found on tooth enamel. These bacteria break down sugar to produce energy and, in the process, cause an acidic environment inside the mouth, which leads to the demineralization of tooth enamel and dental caries [17]. These conditions could lead to early childhood caries (ECC), which is defined as any sign of smooth-surface caries in a child younger than three years of age. ECC can also occur from ages three through five, while one or more cavitated, missing (due to caries), or filled smooth surfaces in the primary maxillary anterior teeth or a decayed, missing, or filled score of greater than or equal to four (age 3), greater than or equal to five (age 4), or greater than or equal to six (age 5) [18].

ECC is the most common infectious disease in young children worldwide [19]. The onset and progression of the disease depend on cultural, social, behavioral, nutritional, and biological risk factors. For example, children who have been exposed to tooth decay in early childhood are more likely to have tooth decay in both deciduous and permanent dentition. ECC not only affects dental health, but also leads to other health consequences such as pain, infection, affecting eating habits, and sleep disorders [19,20].

The essential substrates for cariogenic bacteria are simple carbohydrates such as lactose, sucrose, and glucose. Infants under 12 months of age are fed either breast milk or artificial baby milk; both have about the same content of carbohydrates. In contrast, children older than 12 months, who live in developed countries, usually start drinking cow’s milk at that age, which contains half the amount of carbohydrates contained in breast milk or artificial baby milk [21,22,23]. However, each element is also subject to risk factors such as socioeconomic status, the mother’s education level, maternal oral health, whether the mother smokes and how often, the mother’s posture at birth, the number of sugars in the diet, and oral hygiene and fluoride exposure [23].

Meta-analyses found a reduction in the risk of dental caries in children who breastfed for a longer period of time compared to children who breastfed less. However, infants who breastfed beyond 12 months were at higher risk for tooth decay, especially those who breastfed overnight [23,24].

In conclusion, studies have suggested that breastfeeding for up to 12 months does not involve any increased risk of tooth decay, in fact, it can be a protective factor, compared to artificial baby milk. However, children who have breastfed for more than 12 months, while their entire deciduous teeth have already erupted, have an increased risk of tooth decay. This may be due to other factors associated with unlimited nocturnal feeding. When making the comparison, avoiding additional cariogenic foods and poor oral hygiene habits were taken into account [25].

### 1.3. Objectives

According to the theoretical background, breastfeeding practices have a potentially high impact on the preserved dental hygiene of the infant. A few previous studies have explored the knowledge, attitudes, and practices of mothers toward infant oral health care [26,27,28]. The current study aims to examine the following objectives: (1) Examining the knowledge of breastfeeding mothers about the prevention and health of their infants’ teeth; (2) Examining the socio-demographic factors that are associated with the knowledge regarding prevention and the dental health of toddlers.

## 2. Materials and Methods

### 2.1. Participants and Procedure

We conducted a cross-sectional study design in which breastfeeding mothers completed a questionnaire regarding the health of their toddlers’ teeth. To assess the sample size, we used G-Power software version 27 with the following assumptions: type 1 error of 5% (as the common scientific standard), minimum power required of 80%, and moderate effect size for the correlation between time since birth and knowledge (Pearson correlation of 0.2). Under these assumptions, the required sample size was 153 mothers.

We recruited 165 breastfeeding mothers to participate in this study.

Most of the mothers in this study were employed (73.9%), 14.5% were self-employed, and 11.5% were housewives. Most women had an academic degree at 82.3%, 16.5% were high school graduates, and a small proportion of them (1.2%) had less education than high school. The monthly income of most mothers in this sample (67.3%) ranged between NIS (New Israeli shekels) 5000 and 12,000, meaning an upper–middle-class socioeconomic status.

The participants completed the questionnaires at lactation conferences and public dental clinics. The questionnaires were distributed in 2018 from May to November, at conferences for lactation consultants, in clinics at the Department of Pediatric Dentistry at Tel Aviv University, and at private clinics.

The study was carried out under the supervision of the Institutional Review Board Committee at Tel Aviv University.

### 2.2. Measures and Variables

The questionnaire included three main parts:

A. Socio-demographic background information: Age, employment, education, income, nationality.

B. Birth-related information: Participants were asked to provide information regarding various aspects of the birth, and breastfeeding duration (e.g., toddler’s weight, gestation week at birth).

C. Knowledge regarding the dental health of toddlers: Participants were asked about their knowledge and awareness of the dental health of toddlers (e.g., what are the recommendations for the oral hygiene of toddlers?).

### 2.3. Data Analysis

Data were analyzed using SPSS version 27. First, descriptive statistics for all variables were produced using frequencies (N/%). Since the main variables in this study were measured by categorical responses, we assessed differences between mothers with one child vs. mothers with more than one child using Chi-square procedures. Similarly, we assessed the differences between mothers across education levels using Chi-square procedures. Finally, we assessed the differences between mothers across income levels using Chi-square procedures. The *p*-value for all comparisons was 5%.

## 3. Results

### 3.1. Descriptive Statistics of Breastfeeding Mothers’ Knowledge Regarding Toddlers’ Teeth

According to the socio-demographic information of the sample, about half of the mothers (46.3%) who participated in the study were aged 30–39, 21.3% of the population were aged 20–29, and 32.3% were women over 40 years of age (Table 1).

### 3.2. Knowledge of Breastfeeding Mothers about the Prevention and Health of Their Toddlers’ Teeth

Regarding the mothers’ attitudes toward visiting a pediatric dentist, most of the mothers in the sample (62.1%) were aware of the importance of the first visit to a pediatric dentist and answered that it is recommended to go for a dental examination when the first tooth erupts or from the age of one year (Table 2).

By examining the mothers’ knowledge, we found that 71.2% of mothers responded correctly that the mouth should be cleaned even before the teeth erupt or with the first tooth erupting in the morning and evening, and 80.4% of the sample reported that tooth decay can develop in children under 3. It should be noted that all had statistical significance.

Moreover, 93.2% of the mothers were aware that formula (artificial baby milk) also contained sugar. In addition, about half of the sample (47.8%) were aware that nocturnal feeding may be a risk factor for tooth decay and 48.7% chose the correct answer and answered that bottle-feeding can cause tooth decay even if teeth are brushed.

When asked if the mothers used to taste the food before giving it to their baby, it was found that 62.6% of the mothers used to taste the food before. However, 60.5% answered that they were aware of bacterial transmission from mother to infant.

In another aspect, we examined the mothers’ level of knowledge about the effect of their dental health during pregnancy on their baby’s dental health and found that 68.3% of mothers were unaware of a relationship between their dental health during pregnancy and the baby’s dental health. In addition, 76.2% were unaware of the relationship between the condition of their gums during pregnancy and the possibility of defects and decay in the baby’s teeth.

### 3.3. The Differences between Mothers with One Child in Comparison with Mothers with Multiple Children (Table 3)

By examining the differences between mothers with one child compared with mothers with multiple children, we showed that most mothers (96.8%) with more than one child responded that artificial baby milk includes sugar; in comparison, only 60% of mothers with only one child knew this (*p* < 0.001). In addition, a marginally significant result was found regarding tasting food before giving it to the baby, while most mothers with one child (86.2%) tended to taste in comparison with only 57.9% of the mothers with more than one child (*p* = 0.072).

**Table 3 children-10-00136-t003:** Comparison between mothers with one child vs. multiple children in knowledge about the prevention and health of their toddlers’ teeth.

Question	First Child N ^1^ (%)	Multiple Children N (%)	*p*
**First appointment with a dentist**			0.792
It is recommended to arrive at a dentist when a toddler is 1 y/o	20 (68.9)	62 (65.6)	
It is recommended to arrive at a dentist when a toddler is 3 y/o	9 (31.1)	32 (34.3)	
**Oral hygiene**			0.829
The mouth should be cleaned even before the teeth erupt or when the first tooth erupts in the morning and in the evening.	23 (76.7)	72 (75.8)	
Teeth must be brushed when the first tooth erupts every evening	7 (23.3)	23 (24.2)	
**Do you know if there is sugar in formula (artificial baby milk)?**			<0.001
Formula includes sugar	3 (60.0)	91 (96.8)	
Formula does not include sugar	2 (40.0)	3 (3.2)	
**Association between nocturnal breastfeeding and tooth decay**			0.149
No association	6 (20.7)	29 (31.8)	
Protect from decay	9 (31.3)	13 (14.2)	
Causing a decay	14 (48.0)	49 (53.0)	
**Tasting food before giving it to the baby**			0.072
Yes	25 (86.2)	55 (57.9)	
No	4 (13.8)	40 (42.1)	
**Mother’s caries can infect the baby**			0.872
Yes	18 (62.1)	57 (60.6)	
No	11 (37.9)	37 (39.3)	
**Knowledge regarding the association between a mother’s oral health during pregnancy and a baby’s oral health**			0.034
Yes	12 (40.0)	29 (30.2)	
No	18 (60.0)	67 (69.8)	
**Knowledge regarding the association between maternal periodontal health during pregnancy and the possibility of the baby developing dental problems**			0.024
Yes	10 (33.4)	20 (20.8)	
No	20 (66.6)	76 (79.1)	

^1^ N represents frequencies. (%) Percentage of total is presented in parenthesis.

Regarding knowledge about the association between a mother’s oral health during pregnancy and a baby’s oral health, 40% of mothers with a first child responded positively in comparison with only 30.2% of the mothers with more than one child (*p* = 0.034).

Finally, about one-third (33.4%) of mothers with one child had knowledge regarding a possible association between maternal periodontal health during pregnancy and the possible effect on the baby’s dental health compared with only 20.8% of mothers with more than one child (*p* = 0.024).

### 3.4. Comparison between Mothers According to Education Level

The differences between mothers according to education level were assessed using Chi-square analyses (Table 4).

Examining the differences between mothers with pre-high-school, high school, and academic education in knowledge, it was seen that most mothers (65.9%) with academic education stated that it is recommended to arrive at a dentist when the toddler is 1 y/o, compared with only 50% of mothers with high school education (*p* = 0.002).

In addition, almost all mothers with an academic education (96.7%) responded that formula (artificial baby milk) included sugar in comparison with only 76.9% of mothers with high school education, or 50% with mothers who had a lower education level (*p* < 0.001).

As for tasting food before giving it to their baby, 100% of the mothers with a low education level and 74% of the mothers with high school education tended to taste the food, in comparison with only 60.1% of the mothers with an academic education (*p* = 0.042).

As for the knowledge regarding the association between a mother’s oral health during pregnancy and a baby’s oral health, 50% of the mothers with low education level and 42.3% of mothers with high school education responded positively in comparison with only 29.6% of the mothers with an academic education (*p* < 0.001).

### 3.5. Comparison between Mothers According to Income Levels

The differences between mothers according to income levels were assessed using Chi-square analyses (Table 5).

Examining the differences between mothers according to income levels showed that only a small percentage (27.2%) of low-income mothers stated that it is recommended to arrive a dentist when a toddler is 1 y/o, compared with 67.9% of mothers of moderate income, and 70.3% of mothers with high income (*p* < 0.001).

In addition, higher rates of mothers with moderate income (80.2%) responded that the mouth should be cleaned even before the teeth erupt or when the first tooth erupts in the morning and in the evening, compared with 66.7% of low-income mothers, or 48.1% of high-income mothers (*p* = 0.012).

In our study, 24.8% of mothers breastfed for more than two years, and 48.9%, a large percentage of mothers, breastfed their children at night after the age of one year. In addition, 17.2% breastfed for 3–6 months, 6.7% up to 3 months, and 3.8% breastfed for several days.

## 4. Discussion

The current study examined the knowledge and attitudes of breastfeeding mothers on the dental health of infants and the prevention of early childhood caries. We found several important findings. Our results showed that mothers were found to be highly knowledgeable regarding the oral hygiene of their infants and the recommended breastfeeding best practices. Specifically, most mothers in the current study stated that the first visit to a dentist should be within 6 months of the eruption of the first tooth, and the referral of children to dentists should start at 12 months, as recommended by the American Academy of Pediatrics (AAP) [4]. In addition, most mothers reported that they were aware of the transmission of bacteria from the mother’s mouth to the baby’s mouth, and that it increased the risk of early childhood caries, as found in the literature [26]. However, a large proportion of the mothers in the sample (62%) reported that they usually tasted the food before giving it to their baby, in a way that they could transmit bacteria to infants. Although during pregnancy the mothers’ dental health affects the infants’ dental health, most of the mothers (68%) were not aware of this relationship, and 76.2% were not aware that the mother’s gum condition during pregnancy can affect the development of defects in the baby’s teeth.

We aimed to examine the differences between mothers with a single child in comparison with mothers with several children. Results showed that mothers with more than one child were more knowledgeable regarding the composition of artificial baby milk in comparison with mothers with only a single child. Counterintuitively, most mothers (86%) with more than one child tended to taste the food before giving it to the baby in a way that increases the probability of transmitting bacteria, in comparison with only 58% of the mothers with one child. Mothers with more than one child showed better knowledge regarding the association between the mother’s oral health during pregnancy and the baby’s oral health. These differences demonstrate the experience of mothers with more than one child as they have more knowledge and practice in parenting and the effects of parental practices on the oral hygiene of infants [28]. This result aligns with the distribution of breastfeeding, whilst most of the mothers in our sample breastfed for more than one year (48.9%) or even two years (24.8%).

Examining the association between demographic characteristics with the knowledge and attitudes of the mothers showed that mothers with an academic education demonstrated more knowledge regarding the recommended practices of the infant oral health. This result aligns with the literature that mothers with higher education have a better level of knowledge about their baby’s dental health [20,21,22]. Specifically, mothers with lower education showed poor knowledge regarding the need to refrain from transmitting bacteria by mouth kissing or using the same eating utensil to taste the food before giving it to the baby [26].

According to the literature, the development of early childhood caries involves breastfeeding beyond the first year of the infant’s life. A high frequency of nocturnal breastfeeding, especially after teething, without adherence to oral hygiene care may be an additional risk factor for the development of Severe-ECC [24,29]. In our study, 24.8% of mothers breastfed for more than two years, and 48.9%, a large percentage of mothers, breastfed their children at night after the age of one year. Despite this, the vast majority of mothers 76.6% stated that their children had no or few caries cavities, which indicates relatively good oral health. A possible explanation for this is that mothers, who participated in the study, were adhering to a good dental hygiene care plan, and so despite nocturnal breastfeeding after tooth eruption, dental caries did not arise. This correlates with the fact that the knowledge level regarding when to start brushing teeth, and when to go for the first dental checkup, was high.

Our results should be considered according to the following limitations. First, the cohort was relatively small (N = 165) and data were gathered using a convenience sampling method. Therefore, it is important that future studies will replicate this study using a larger and more representative sample. Second, although our study focused on breastfeeding mothers, the results were relatively limited, since our design did not include a control group of non-breastfeeding mothers. This group should provide an understanding of which knowledge and attitudes are indeed unique for breastfeeding mothers compared to non-breastfeeding mothers.

Our studies contribute to the literature by stressing the importance of providing professional knowledge to breastfeeding mothers in order to improve the oral health of their children from birth. This conclusion is in line with recent studies that emphasized the need to approach breastfeeding mothers with regard to enhancing their knowledge and practices about the oral hygiene of their infants [30,31,32].

To conclude, this study examined the knowledge of breastfeeding mothers about caries prevention and the health of their infants’ teeth. In addition, we examined the socio-demographic factors that are associated with the knowledge regarding the prevention and dental health of infants. 

Overall, we present here an exploratory study that provides rich information regarding the caries prevention knowledge of breastfeeding mothers. Hence, we used descriptive statistics for various aspects of the mothers’ knowledge. 

Although according to the results of the present work the nursing mothers showed good knowledge, there is a lack of knowledge and a wrong attitude in some aspects that need to be improved:Knowledge of breastfeeding mothers with lower education about their baby’s dental health;Knowledge and attitudes of breastfeeding mothers with only one child;Awareness of the risk of transferring bacteria to the baby’s mouth by tasting the food. Thus, they may increase the risk of dental caries;Awareness of breastfeeding mothers that their dental health during pregnancy affects the infants’ dental health.

The results of this study shed light on the possibility that maternal knowledge regarding dental hygiene and caries prevention through timely toddler dental appointments could be a game-changer in the dental health of breastfeeding toddlers. This important aspect could serve policymakers in improving practices toward advancing better oral health for infants while preserving breastfeeding. Educating mothers about these two relatively simple practices, starting good oral hygiene even before the first tooth erupts and a timely first dental appointment when the toddler is one year of age, can prevent dental caries in breastfeeding toddlers.

## Figures and Tables

**Table 1 children-10-00136-t001:** Socio-demographic information of the sample (n = 165).

Variable	N ^1^	%
**Age** (years)		
20–29	35	21.3
30–39	76	46.3
≥40	53	32.3
**Employment**		
Housewife	19	11.5
Employee	122	73.9
Self-employed	24	14.5
**Education**		
Lower than high school	2	1.2
High school	27	16.5
Academic	136	82.3
**Income (NIS ^2^)**		
<5000	26	15.7
5000–12,000	111	67.3
>12,000	28	17.0
**Number of children**		
1	33	19.5
2	59	36.0
3 or more	73	44.5
**Age of breastfed child**		
1–1.5	53	43.4
1.5–2	22	18.0
2 or older	47	38.5

**Table 2 children-10-00136-t002:** Knowledge of breastfeeding mothers about the prevention and health of their toddlers’ teeth.

Question	N ^1^	%
**First appointment with a dentist**		
*It is recommended to arrive at a dentist when a toddler is 1 y/o?*	100	62.1
It is recommended to arrive at a dentist when a toddler is 3 y/o?	61	37.9
**Oral hygiene**		
*The mouth should be cleaned even before the teeth erupt or when the first tooth erupts in the morning and in the evening.*	116	71.2
Teeth must be brushed when the first tooth erupts every evening	47	28.8
**Development of tooth decay in children under the age of 3**		
*Decay could develop*	131	80.4
Decay could not develop	32	19.6
**Do you know if there is sugar in Formula (artificial baby milk)?**		
*Formula includes sugar*	150	93.2
Formula does not include sugar	11	6.8
**Association between nocturnal breastfeeding and tooth decay**		
No association	54	34.4
Protect from decay	28	17.8
*Causing a decay*	75	47.8
**Effect of bottle feeding on caries development**		
Bottle-feeding could lead to the development of tooth decay only if the toddler does not brush their teeth	80	51.3
*Caries can be caused by bottle feeding even if a toddler brushes his teeth*	76	48.7
**Tasting food before giving it to the baby**		
Yes	102	62.6
*No*	61	37.4
**Mother’s caries can infect the baby**		
*Yes*	98	60.5
No	64	39.5
**Knowledge regarding the association between a mother’s oral health during pregnancy and a baby’s oral health**		
*Yes*	52	31.7
No	112	68.3
**Knowledge regarding the association between maternal periodontal health during pregnancy and the possibility of the baby developing dental problems**		
*Yes*	39	23.8
No	125	76.2

^1^ N represents frequencies.

**Table 4 children-10-00136-t004:** Comparison between mothers according to education levels in knowledge about caries prevention and the health of their toddlers’ teeth.

Question	Lower Than High School N ^1^ (%)	High School N (%)	Academic Education N (%)	*p*
**First appointment with a dentist**				0.002
It is recommended to arrive at a dentist when a toddler is 1 y/o?	0	13 (50.0)	87 (65.9)	
It is recommended to arrive at a dentist when a toddler is 3 y/o?	2 (100.0)	13 (50.0)	45 (34.1)	
**Oral hygiene**				0.129
The mouth should be cleaned even before the teeth erupt or when the first tooth erupts in the morning and in the evening.	1 (50.0)	14 (53.8)	100 (74.6)	
Teeth must be brushed when the first tooth erupts every evening	1 (50.0)	12 (46.2)	34 (25.4)	
**Do you know if there is sugar in Formula (artificial baby milk)?**				<0.001
Formula includes sugar	1 (50.0)	20 (76.9)	128 (96.7)	
Formula does not include sugar	1 (50.0)	6 (23.1)	4 (3.3)	
**Association between nocturnal breastfeeding and tooth decay**				0.189
No association	0	9 (34.6)	44 (33.8)	
Protect from decay	0	4 (15.4)	24 (18.4)	
Causing a decay	0	13 (50.0)	62 (47.7)	
**Effect of bottle feeding on caries development**				0.241
Bottle-feeding could lead to the development of tooth decay only if the toddler does not brush the teeth	1 (100.0)	17 (65.3)	61 (47.6)	
Caries can be caused by bottle feeding even if a toddler brushes the teeth	0	9 (34.7)	67 (52.4)	
**Tasting food before giving it to the baby**				0.042
Yes	2 (100.0)	20 (74.0)	80 (60.1)	
No	0	7 (26.0)	53 (39.9)	
**Mother’s caries can infect the baby**				0.281
Yes	1 (50.0)	14 (51.8)	83 (62.8)	
No	1 (50.0)	13 (48.2)	49 (37.2)	
**Knowledge regarding the association between a mother’s oral health during pregnancy and a baby’s oral health**				<0.001
Yes	1 (50.0)	11 (42.3)	40 (29.6)	31.7
No	1 (50.0)	15 (57.7)	95 (70.3)	68.3
**Knowledge regarding the association between maternal periodontal health during pregnancy and the possibility of the baby developing dental problems**				0.291
Yes	1 (50.0)	7 (26.9)	31 (22.9)	23.8
No	1 (50.0)	19 (73.1)	104 (77.1)	76.2

^1^ N represents frequencies. (%) Percentage of total is presented in parenthesis.

**Table 5 children-10-00136-t005:** Comparison between mothers according to income levels in knowledge about caries prevention and the health of their toddlers’ teeth.

Question	NIS < 5000 ^2^ N ^1^ (%)	NIS 5000–12,000 N (%)	NIS > 12,000 N (%)	*p*
**First appointment with a dentist**				<0.001
It is recommended to arrive at a dentist when a toddler is 1 y/o	6 (27.2)	72 (67.9)	19 (70.3)	
It is recommended to arrive at a dentist when a toddler is 3 y/o	16 (72.8)	34 (32.1)	8 (29.7)	
**Oral hygiene**				0.012
The mouth should be cleaned even before the teeth erupt or when the first tooth erupts in the morning and in the evening.	16 (66.7)	85 (80.2)	13 (48.1)	
Teeth must be brushed when the first tooth erupts every evening	8 (33.3)	21 (19.8)	14 (51.9)	
**Do you know if there is sugar in artificial baby milk**				0.581
Artificial baby includes sugar	20 (83.4)	100 (95.2)	25 (92.6)	
Artificial baby does not include sugar	4 (16.6)	5 (4.8)	2 (7.4)	
**Association between nocturnal breastfeeding and tooth decay**				0.763
No association	6 (26.1)	37 (36.3)	9 (34.6)	
Protect from decay	5 (27.1)	18 (17.6)	4 (15.4)	
Causing a decay	12 (52.2)	47 (46.1)	13 (50.0)	
**Effect of bottle feeding on caries development**				0.424
Bottle-feeding could lead to the development of tooth decay only if the toddler does not brush their teeth	16 (66.7)	50 (49.5)	11 (42.3)	
Caries can be caused by bottle feeding even if a toddler brushes the teeth	8 (33.3)	51 (50.5)	15 (57.7)	
**Tasting food before giving it to the baby**				0.628
Yes	15 (62.5)	69 (65.1)	13 (48.1)	
No	9 (37.5)	37 (34.9)	14 (51.9)	
**Mother’s caries can infect the baby**				0.579
Yes	14 (58.3)	65 (61.9)	16 (59.3)	
No	10 (41.7)	40 (38.1)	11 (40.7)	

^1^ N represents frequencies. (%) Percentage of total is presented in parenthesis. ^2^ NIS, New Israeli Shekel.

## Data Availability

All data supporting the reported results can be made available upon reasonable request from the corresponding author.

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
