# Peer review of "Knowledge of Breastfeeding Mothers Regarding Caries Prevention in Toddlers"

_children, 2023, doi:10.3390/children10010136_

Round 1

Reviewer 1 Report

The article "Knowledge of Breastfeeding Mothers regarding Caries Prevention in toddlers" reports the results of a survey conducted among breastfeeding mothers in Tel-Aviv. Socio-demographic factors associated with the knowledge on the oral health of toddlers were explored. The article is well-written and interesting for the readers. Also, it is devoted to an important topic. I think that it can be published after a few minor corrections.

First, please, check once again spelling and grammar.

Second, please, provide the information in the M&Ms section on the size of the target population that was used to calculate the study sample size. 

Author Response

Dear reviewer,

Thank you for reviewing our article and for your constructive notes.

  1-We checked again and corrected spelling and grammar. Attached is a corrected manuscript.

2-The sample size calculation section was elaborated.

. To assess sample size, we used G-Power software with the following assumptions

            Type 1 error of 5% (as the common scientific standard) a required statistical power of 80%.

                        A moderate effect size, based on the expected correlation between time since birth and Knowledge regarding oral health hygiene (Pearson correlation of 0.2)

Under these assumptions, the required sample size is 153 mothers.

Reviewer 2 Report

Thank you for allowing me to review this scientific paper that examined breastfeeding mothers' knowledge of their baby's dental health and prevention of ECC.

The article has an interesting topic, but there are areas that can be improved in implementation. It is a cross-section of a study that should follow the guidelines of STROBE in writing it, which this article does not. The methodology is flawed and does not meet the requirements for scientific work.

These are just some of the suggestions for improving the article by correcting the English language.

1. The first sentence of the abstract is clear, and why knowledge of mothers is important.

3. Present the results numerically in the abstract. More precisely, could you support them with numbers?

3. Correct the last sentence of the first paragraph of the introduction and the second sentence of the second paragraph.

4. Please explain more clearly the calculation of the sample size.

5. Table 1 from the methods, according to the results with the corresponding text.

6. Who designed the questionnaire, how many questions they had, what were the questions, who validated it and how? What design were the questions asked?

7. Where was the research conducted, in what period, and what were the inclusion and exclusion criteria?

8. How was the research conducted live or online?

9. How was the statistics conducted, with which package, and with which methods - please describe in more detail.

10. In Table 2, mark the correct answers with a superscript. How was knowledge assessed, based on the sum of correct answers or otherwise? Why is the total knowledge score not correlated with the respondents' demographic data, but it was done with the chi-square test? You have chosen the wrong statistical method. Literally, from table 2 to table five, similar results could be obtained by regression in relation to the knowledge score.

11. What is the strength of this study?

Author Response

Thank you for reviewing our study

Below are our responses to your comments:

  1. The first sentence of the abstract is clear, and why knowledge of mothers is important.

                              Please clarify what you mean

  1. Present the results numerically in the abstract. More precisely, could you support them with numbers?

corrected

  1. Correct the last sentence of the first paragraph of the introduction and the second sentence of the second paragraph.

corrected

  1. Please explain more clearly the calculation of the sample size.

                       The sample size calculation section was elaborated.

  1. Table 1 from the methods, according to the results with the corresponding text.

corrected

  1. Who designed the questionnaire, how many questions they had, what were the questions, who validated it and how? What design were the questions asked?

                     The questionnaire was based on a previous article that used a similar study: reference 26

  There were 41 questions. two parts in the questionnaire:

   a-demographic data 

    b- Knowledge of breastfeeding mothers about dental health and prevention of tooth

               decay and early childhood, and the Eating and breastfeeding habits of the baby 

  1. Where was the research conducted, in what period, and what were the inclusion and exclusion criteria?

            The questionnaires were distributed in 2018 from May to November.

              At conferences for lactation consultants, in clinics in the Department of Pediatric   

           Dentistry at Tel Aviv University, and in private clinics.

Added to the methods

8 . How was the research conducted live or online?

          Live

9  How was the statistics conducted, with which package, and with which methods - please describe in more detail.

     The data analysis section was elaborated.

  1. In Table 2, mark the correct answers with a superscript. How was knowledge assessed, based on the sum of correct answers or otherwise? Why is the total knowledge score not correlated with the respondents' demographic data, but it was done with the chi-square test? You have chosen the wrong statistical method. Literally, from table 2 to table five, similar results could be obtained by regression in relation to the knowledge score.

The knowledge was assessed using the sum of correct answers.

Intentionally we aim to show the specific questions and answers, and their correlations with the main demographic variables. Hence, we did not choose regressions, but rather Chi-square tests.

11 What is the strength of this study?

         According to the results of our study the knowledge of mothers regarding the prevention and

        oral health of their toddlers is not enough, especially for mothers with low income and  

            lower education.

        Most mothers are not aware of the effect of prolonged breastfeeding on their toddler's oral 

               health. Therefore, it is very important that pregnant mothers be aware of the possible

                    relationship between breastfeeding and dental caries.

Round 2

Reviewer 2 Report

The authors did not numerically list the results in the abstract. The conclusion in the abstract is too general and is not based on the obtained results.

On the basis of which data was the strength of the study calculated, this research, other research, pilot study?

Demographic data does not belong to the methodology but to the results. The authors stated that they corrected the same, but they did not (Table 1).

In Table 2, please mark the correct answers in italics.

Combine Table 2 and 3 into one.

In different tables, the authors repeat similar results depending on demographics, which could be tested by correlation.

Also, the conclusion of the artice is subjective and not based on the results obtained.

Author Response

Thank you for your helpful notes.

Here are our responses:

The authors did not numerically list the results in the abstract.

The conclusion in the abstract is too general and is not based on the obtained results.

Corrected. I added the results with numbers and changed the conclusion to be more specific.

On the basis of which data was the strength of the study calculated, this research, other research, pilot study?

Based on pilot study

Demographic data does not belong to the methodology but to the results.

Corrected. The demographic data was added to the results. (first paragraph)

In Table 2, please mark the correct answers in italics.

Done.

Combine Tables 2 and 3 into one.

We intentionally separated the two tables because we wanted to emphasize two facts: 1- knowledge of mothers regardless of the number of children.

2- Comparison of knowledge and attitude between mothers with one child and mothers with more than one child. This was one of the aims of the study.

Also, the conclusion of the article is subjective and not based on the results obtained.

Elaborated:

Although according to the results of the present work, the nursing mothers showed good knowledge, there is a lack of knowledge and a wrong attitude in some aspects that need to improve:

1- Knowledge of breastfeeding mothers with lower education about their baby's dental health.

2 Knowledge and attitudes of breastfeeding mothers with only a single child.

3- Awareness of the risk of transferring bacteria to the baby's mouth by tasting the food. Thus, they may increase the risk of dental caries.

4- awareness of breastfeeding mothers that their dental health during pregnancy affects the infants’ dental health.

 The results of this study shed light on the knowledge of breastfeeding mothers on this important topic and could serve policymakers to improve practices towards advancing better oral health for infants, without sacrificing the benefits of breastfeeding, which are so crucial for infant health and development.